# Nef mediates neuroimmune response, myelin impairment, and neuronal injury in EcoHIV-infected mice

Jessica K Schenck⬥, Cheryl Clarkson-Paredes, Tatiana Pushkarsky, Yongsen Wang, Robert H Miller⬥,
Michael I Bukrinsky⬥

**The introduction of antiretroviral therapy has markedly improved the management of HIV-associated neurocognitive disorders (HAND). However, HAND still affects nearly half of HIV-infected individuals, presenting significant challenges to their well-being. This highlights the critical need for a deeper understanding of HAND mechanisms. Among HIV viral proteins, Nef is notable for its multifaceted role in HIV pathogenesis, though its specific involvement in HAND remains unclear. To investigate this, we used a murine model infected with Nef-expressing (EcoHIV) and Nef-deficient (EcoHIVΔNef) murine HIV. Comparative analyses revealed increased neuroinflammation and reduced myelin and neuronal integrity in EcoHIV-infected brains compared with those with EcoHIVΔNef. Both viruses induced astrogliosis, with stronger GFAP activation in Nef-deficient infections. These findings suggest that Nef contributes to neuroinflammation, primarily through microglial targeting and demyelination, although other factors may regulate astrogliosis. Our results indicate that Nef may significantly contribute to neuronal injury in EcoHIV-infected mice, offering insights into Nef-induced neuropathology in HAND and guiding future research.**

## Introduction

Despite achieving sustained virologic control with antiretroviral therapy (ART), up to half of all people living with HIV (PLWH) remain afflicted by HIV-associated neurocognitive disorders (HAND) (1, 2, 3). The pathogenesis of HAND and its relationship to HIV infection, particularly in the era of ART, however, remain incompletely understood. Chronic neuroinflammation is widely believed to be a central aspect of HAND pathogenesis (4), yet the origins of HIV-induced neuroinflammation and the mechanisms that result in the persistence of inflammation in the brains of aviremic individuals remain elusive.

A promising candidate HIV protein implicated in neuroinflammation is Nef. During systemic HIV infection, Nef triggers robust inflammatory responses in the periphery by suppressing ATP-binding cassette A1 (ABCA1)-mediated cholesterol efflux and augmenting lipid raft abundance in myeloid cells (5, 6). Within the brain, Nef is synthesized by infected microglia and astrocytes, even during successful ART therapy (7, 8, 9), and has been associated with HAND pathogenesis (10, 11, 12). Nef has been shown to disrupt the proper differentiation of oligodendrocytes in vitro (13), potentially contributing to the persistent white matter damage observed in the ART era (14, 15). Furthermore, recent animal studies have demonstrated that astrocytic expression of Nef induces deficits in spatial and recognition memory, correlating with neuroinflammation (16, 17).

Although insights into underlying drivers of HIV neuropathology can be gleaned from cerebrospinal fluid sample collection, imaging studies, and postmortem analysis of human brain samples, experimental approaches to elucidate disease mechanisms and identify therapeutic targets are not feasible in humans. Mice serve as a predominant animal model for studying neurological disorders due to the ability to manipulate their genome and the similarities between mice and humans in the process of neuronal degeneration and demyelination observed in neurodegenerative diseases such as Alzheimer's, multiple sclerosis, and Parkinson's (18). Although conventional mice are naturally resistant to HIV infection, the EcoHIV model, a chimeric murine HIV developed by Potash et al (19), circumvents this limitation. In EcoHIV, HIV tropism is conferred in mice by replacing the HIV-1 gp120 envelope gene with the murine leukemia virus ecotropic gp80 envelope gene (19). EcoHIV infection mirrors many critical aspects of HIV-1 infection in humans on ART (reviewed in reference 20). It targets CD4⁺ T lymphocytes, macrophages, and microglia, establishing reservoirs with persistent integrated replication-competent virus (21). Similar to aviremic patients on stable ART, EcoHIV-infected mice maintain stable viral suppression without progressing to immunodeficiency (21). EcoHIV also exhibits neuroinvasion, spreading to brain tissue in systemically infected mice (19, 21). This CNS infection correlates with neurocognitive impairment, non-apoptotic hippocampal synaptodendritic injury, and elevated inflammatory responses in brain tissue despite intact host immunity (20, 21).

In this study, we aimed to elucidate the role of Nef in neuroinflammation and associated brain pathologies observed in

---

School of Medicine and Health Sciences, The George Washington University, Washington, DC, USA

Correspondence: mbukrins@gwu.edu

 

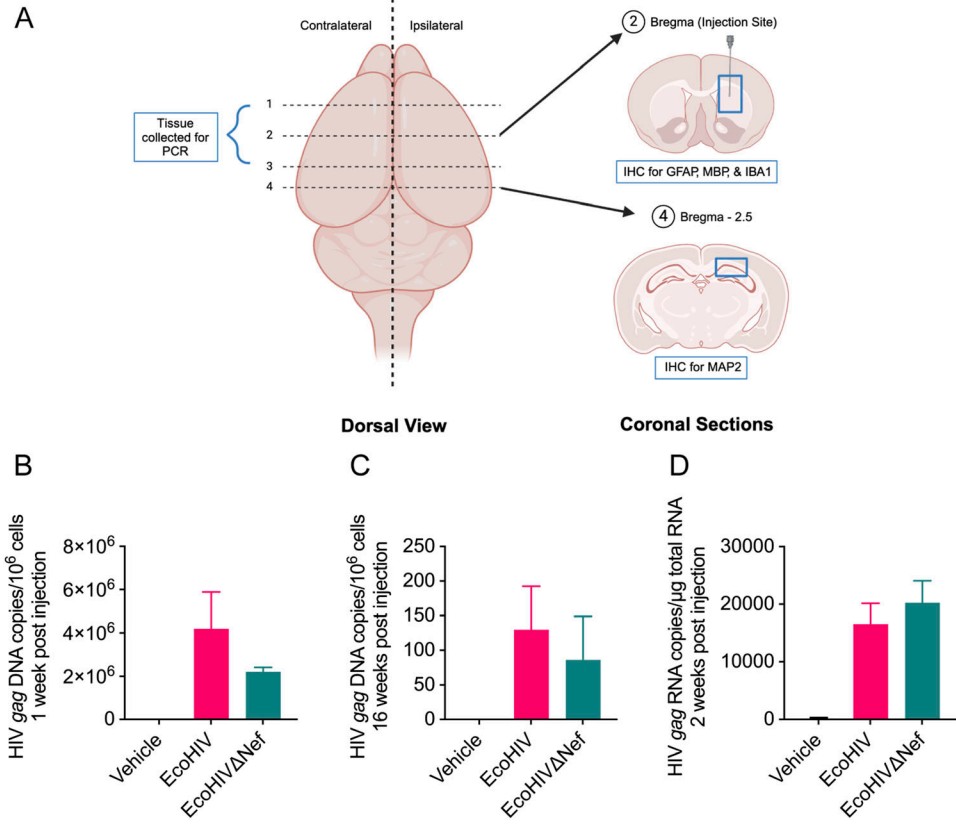

**Figure 1. Intracranial infection with EcoHIV and EcoHIVΔNef.**
Mice were intracranially injected with vehicle (mock-infected), EcoHIV, or EcoHIVΔNef.
**(A)** Schematic depicting the mouse brain, including the site of injection and location for molecular and immunohistochemical analyses. Image created in BioRender.
**(B)** Quantification of HIV *gag* DNA from 3 mm sections of the ipsilateral hemisphere at the injection site 1-wk post infection by qPCR (n = 3). **(C)** Quantification of HIV *gag* DNA from 3 mm sections of the ipsilateral hemisphere 16-wk post infection by qPCR (n = 2).
**(D)** Quantification of HIV *gag* RNA from 3 mm sections of the ipsilateral hemisphere at the injection site 2-wk post infection by qRT-PCR (n = 4). Results are presented as mean ± SEM.

EcoHIV-infected mice. To achieve this objective, we used a Nef-deficient EcoHIV (EcoHIVΔNef) constructed by removing a portion of the Nef coding region from the EcoHIV virus. Generating Nef-deleted or Nef-deficient constructs is a common methodology for examining the effects of Nef and has been used to study the effects of Nef in the context of HIV infection in vitro and ex vivo as well as in simian immunodeficiency virus-infected Rhesus macaques (22, 23, 24, 25). We intracranially injected mice with EcoHIV and EcoHIVΔNef to examine the contribution of Nef in EcoHIV-associated neuropathology. Our results demonstrate the essential role of Nef in mediating inflammatory responses and brain damage associated with EcoHIV infection.

## Results

### EcoHIV and EcoHIVΔNef establish infection in mouse brain

Although Nef is dispensable for HIV-1 replication in vitro, it markedly elevates viral replication in vivo (26, 27). EcoHIV DNA and RNA have previously been found in brain tissues of intracranially inoculated mice up to 12 wk post injection (28) and in the spleens and peritoneal cells of peripherally inoculated mice up to 450 d post injection (29). Previous reports have indicated that EcoHIV is localized to IBA1+ microglia/macrophages in the brain in vivo (29) and in mixed brain cultures in vitro (30). To confirm that EcoHIV and EcoHIVΔNef can infect brain tissue in our model, we

injected adult C57BL/6J mice (Fig 1A) with vehicle, EcoHIV, or EcoHIVΔNef and measured HIV burden in brain tissue within the ipsilateral (injected) hemisphere at the injection site at 1-, 2-, and 16-wk post infection by real-time quantitative PCR (qPCR). There was a small and nonsignificant decrease in HIV *gag* DNA copy number in EcoHIVΔNef-compared with EcoHIV-infected mice at 1-wk post infection (Fig 1B); this was supported by analysis at 16-wk post infection (Fig 1C). The difference in HIV *gag* RNA expression between EcoHIVΔNef- and EcoHIV-infected mice at 2-wk post infection was also minimal (Fig 1D). These results indicate effective replication and stable persistence of EcoHIVΔNef virus in infected mice, comparable to that of EcoHIV. For both EcoHIV- and EcoHIVΔNef-infected mice, HIV *gag* DNA was detected in brain tissue from the ipsilateral (injected) hemisphere, although negligible viral DNA was detected in the contralateral hemisphere (not shown), suggesting that integration and replication for both viruses was localized to the site of injection.

### Nef mediates neuroinflammation in EcoHIV-infected mice

Previous studies reported that EcoHIV-infected mouse brains display increased activation and infiltration of IBA1+ microglia and show up-regulation of inflammation-related genes associated with HAND that peak at 2-wk post infection (31). We have previously shown that Nef EVs induce hyperreactivity of myeloid cells (6), which may contribute to the persistent inflammation and associated pathologies observed in stably suppressed HIV-infected

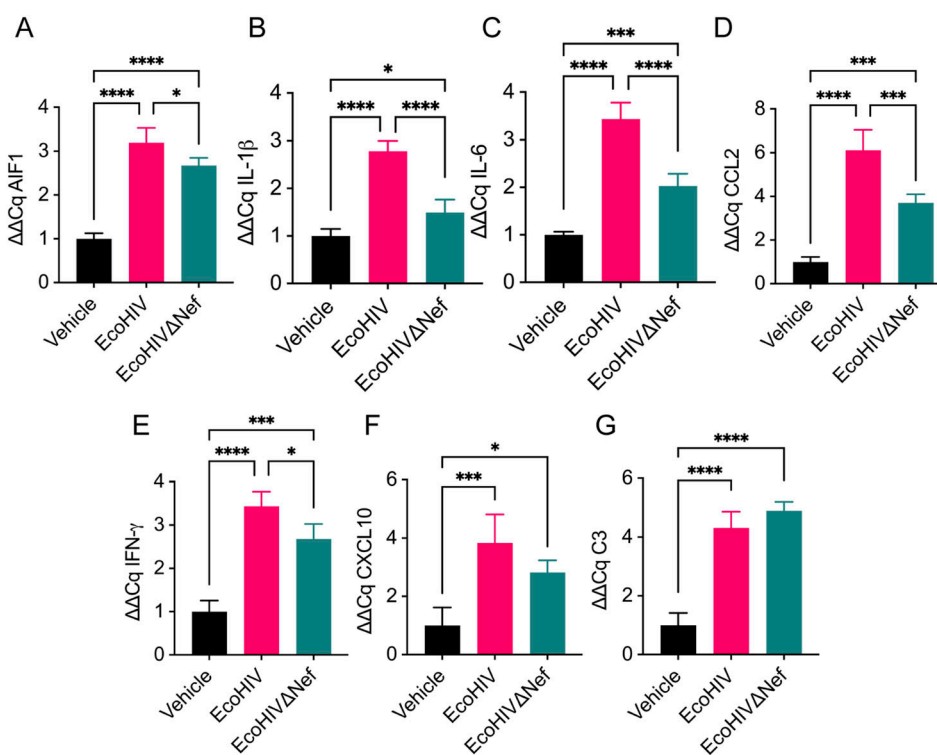

**Figure 2. Nef mediates expression of pro-inflammatory genes in EcoHIV-infected mice.**
Mice were intracranially injected as in Fig 1. At 2-wk post injection, brain tissue was collected from 3 mm sections of ipsilateral hemisphere at the injection site and total cellular RNA was analyzed by qRT-PCR. **(A)** Analysis of AIF1. **(B)** Analysis of IL-1$\beta$. **(C)** Analysis of IL-6. **(D)** Analysis of CCL2. **(E)** Analysis of IFN-$\gamma$. **(F)** Analysis of CXCL10. **(G)** Analysis of C3. Results are shown as mean fold change ± SD normalized to vehicle (set as 1.0) and analyzed by ANOVA. n = 4 mice per group. *$P$ < 0.05; ***$P$ < 0.001; ****$P$ < 0.0001.

individuals on ART. To determine if Nef contributes to persistent neuroinflammation in EcoHIV-infected mice, we compared samples from animals infected with EcoHIV or EcoHIVΔNef. RNA extracted from 3 mm coronal sections of flash frozen brain tissue at the injection site 2-wk post infection was assayed for expression of pro-inflammatory genes associated with HAND. In parallel, 10 $\mu$m fixed brain sections were analyzed by immunohistochemistry for expression of IBA1 at 5-wk post infection, well after the peaks of HIV and pro-inflammatory transcript expression seen at 1- to 2-wk post infection.

Consistent with previous reports, qPCR analysis of EcoHIV-infected brains 2-wk post infection revealed a significant 2.8-fold to 6.1-fold up-regulation of several pro-inflammatory genes relative to vehicle injected mice. Up-regulated genes included the macrophage/microglial marker AIF1 (IBA1) (Fig 2A), cytokines IL-1$\beta$ (Fig 2B) and IL-6 (Fig 2C), and the chemokine CCL2 (MCP-1) (Fig 2D). The up-regulation of these genes was notably mitigated in Eco-HIVΔNef-infected brains compared with those infected with EcoHIV (Fig 2A–D), although still elevated relative to vehicle-injected controls. These findings underscore the pivotal role of Nef in eliciting these pro-inflammatory responses, although other viral factors likely play contributory roles.

The expression of IFNs, a class of antiviral cytokines that play a key role in the immune response to viral infections (32), was also up regulated by HIV infection. IFN-$\gamma$ was significantly up-regulated in EcoHIV-infected mice compared with both mock-infected and EcoHIVΔNef-infected animals (Fig 2E). The IFN-$\gamma$ inducible protein CXCL10 (IP-10) followed a similar pattern, showing significant up-regulation in EcoHIV-infected mice relative to mock-infected controls, although the difference between EcoHIV-infected and

EcoHIVΔNef did not reach statistical significance (Fig 2F). This suggests that Nef contributes to the induction of IFN-responses in EcoHIV-infected brains. Not all inflammatory factors were affected by Nef. For example, complement component C3 was equally elevated in both EcoHIV- and EcoHIVΔNef-infected mice, suggesting that viral mechanisms other than Nef contribute to the C3 response (Fig 2G).

Immunohistochemical analysis of the basal ganglia at the injection site 5-wk post infection revealed heightened microglia/myeloid cell activation and increased numbers of cells in mice infected with EcoHIV compared with EcoHIVΔNef-infected or mock-infected animals (Fig 3A). Quantitative assessment of IBA1 expression showed an 86% increase in IBA1 staining in EcoHIV compared with mock-infected counterparts, and a 36% increase relative to EcoHIVΔNef-infected samples, corresponding to an overall rise in IBA1-positive cell density (83% increase relative to mock-infected, 58% increase relative to EcoHIVΔNef-infected) (Fig 3B and C). These findings underscore the pivotal role of Nef in mediating persistent microglial cell activation.

Astrogliosis is another characteristic feature of neuroinflammation. In previous studies, IHC analysis of EcoHIV-infected mice showed an increase in GFAP staining in the basal ganglia in the vicinity of the injection site at 2-wk post infection (33). By contrast, other studies have reported that Nef expression leads to down-regulation of the GFAP gene expression in astrocytes in vitro (34), and we observed that injection with Nef-containing extra-cellular vesicles results in decreased density of GFAP+ cells within the corpus callosum 3-d post injection (35). To determine whether EcoHIV disrupts white matter astrocytes in a Nef-mediated mechanism, we examined expression of GFAP within the corpus

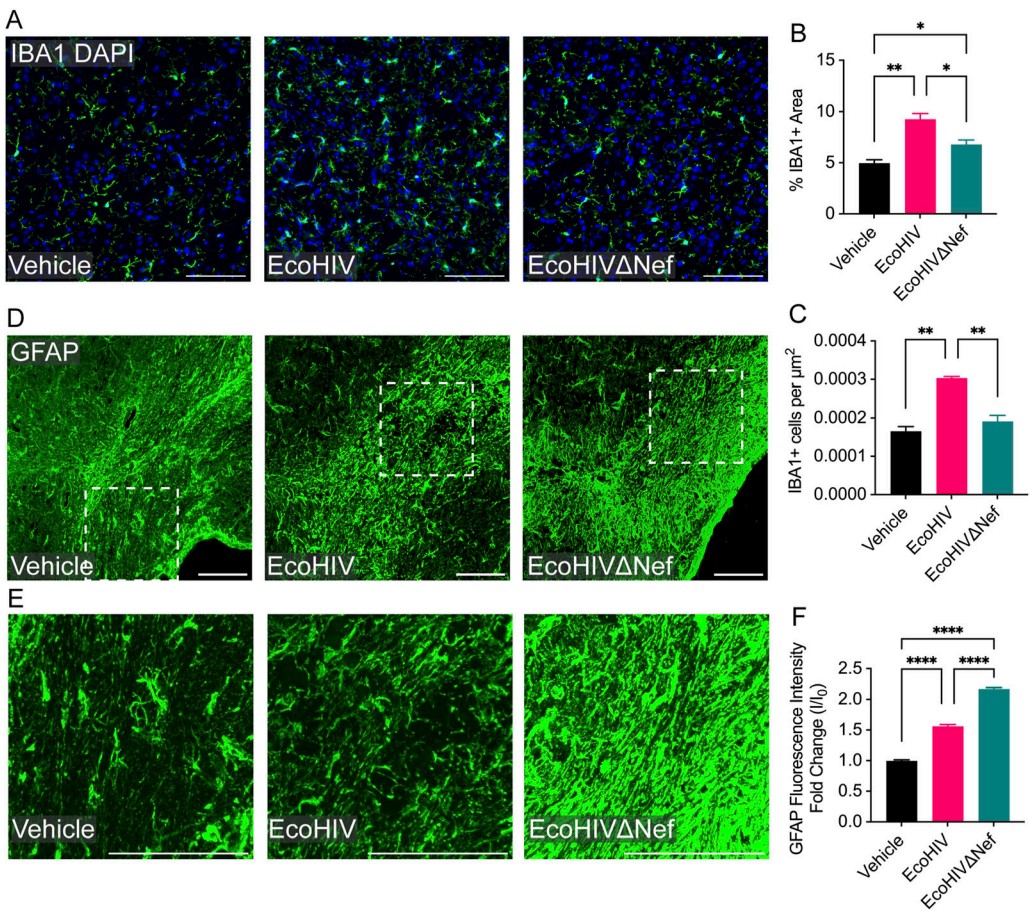

**Figure 3. EcoHIV induces activation of microglia and astrocytes.**
EcoHIV induces activation of microglia and astrocytes. Mice were intracranially injected as in Fig 1. At 5-wk post infection, brain tissue was processed for analysis by immunohistochemistry. Images show representative fields from the injection site. **(A)** Representative fluorescence images of microglia/macrophages (IBA1, green) and nuclei (DAPI, blue) from the basal ganglia. **(B)** Quantification of % IBA1+ immunofluorescence area within the basal ganglia at the injection site. **(C)** Quantification of IBA1+ cells per μm2 within the ipsilateral basal ganglia, n = 2–4 mice per group. **(D)** Representative fluorescence images of astrocytes (GFAP, green) at the corpus callosum. **(D, E)** Higher magnification images of the cells within insets in panel (D). **(F)** Quantification of GFAP fluorescence intensity, n = 3–4 mice per group. Scale bars in all figures are 100 μm. Results are shown as mean ± SEM. IBA1 results were analyzed by ANOVA and GFAP by Kruskal-Wallis rank sum test. *P < 0.05; **P < 0.01; ****P < 0.0001.

callosum at the injection site 5-wk post infection via confocal microscopy (Fig 3D). In line with previous experiments examining the effects of EcoHIV on gray matter astrocytes (33), EcoHIV-infected mice displayed increased GFAP fluorescence intensity within the corpus callosum compared with vehicle-injected mice, reflecting EcoHIV-mediated astrogliosis, which was further elevated in EcoHIVΔNef-infected mice (Fig 3E). Quantification of these changes demonstrated a 56% increase of GFAP fluorescence intensity in EcoHIV-infected relative to uninfected control mice, a 40% increase in EcoHIVΔNef-infected relative to EcoHIV-infected mice, and a 118% increase in EcoHIVΔNef-infected relative to vehicle-injected mice (Fig 3F), suggesting that the presence of Nef had a significant effect on the suppression of the astrocyte reactive response. We conclude that EcoHIV infection induces a strong astrogliosis response and this response is mitigated by the presence of Nef that mediates down-regulation of GFAP expression in white matter astrocytes. Taken together, the results in Figs 2 and 3 indicate that Nef contributes to specific inflammatory pathologies in EcoHIV-infected mice.

## Nef-mediated impairment of ABCA1 and triggering receptor expressed on myeloid cells 2 (TREM2) expression in EcoHIV-infected mice

During systemic HIV infection, Nef triggers inflammatory responses by suppressing ABCA1-mediated cholesterol efflux and increasing lipid raft abundance on myeloid cells (5, 6). Another regulator of lipid metabolism in microglia is TREM2 (36) that is linked to CNS HIV disease severity (37). We investigated whether ABCA1 and TREM2 expression in brain tissues of EcoHIV-infected mice was disrupted and influenced by the presence of Nef. Mice were injected with EcoHIV or EcoHIVΔNef, and RNA from brain sections at the injection site was analyzed 2 wk post infection. ABCA1 RNA was significantly down-regulated in both EcoHIV- and EcoHIVΔNef-infected mice (Fig 4A), although TREM2 expression was significantly up-regulated (Fig 4B), indicating disruption of these genes associated with CNS lipid transport and metabolism. No apparent differences in ABCA1 and TREM2 transcript expression were observed between EcoHIV- and EcoHIVΔNef-infected mice, suggesting that Nef did not contribute to

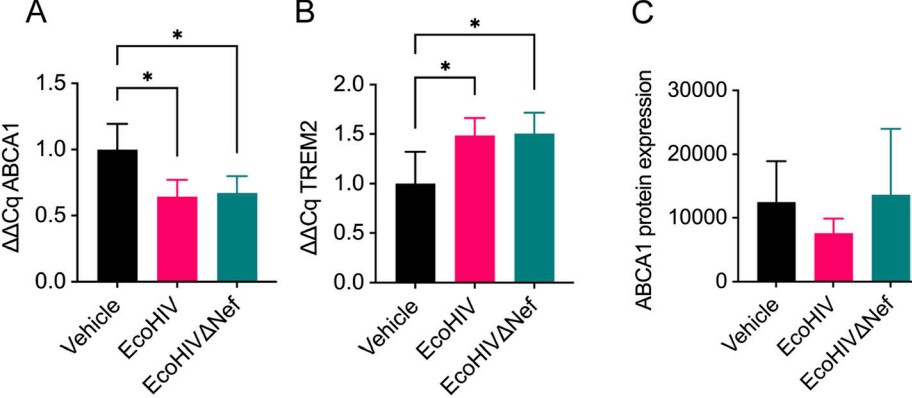

**Figure 4.   EcoHIV infection affects brain ABCA1 and TREM2 expression.**
Mice were intracranially injected as in Fig 1. **(A, B)** At 2-wk post infection, brain tissue was collected from 3 mm sections of ipsilateral hemisphere at the injection site and total cellular RNA was analyzed by qRT-PCR for ABCA1 (A) and TREM2 (B). Results are shown as mean fold change ± SD normalized to vehicle (set as 1.0) and analyzed by ANOVA. n = 4 mice per group. *$P < 0.05$. **(C)** At 5-wk post injection, brain tissue was collected and total ABCA1 protein was analyzed within the ipsilateral hemisphere via capillary electrophoresis. Results are presented as mean ± SEM with ABCA1 protein abundance adjusted to total protein abundance and analyzed by Kruskal-Wallis rank sum test. n = 2–3 mice per group.

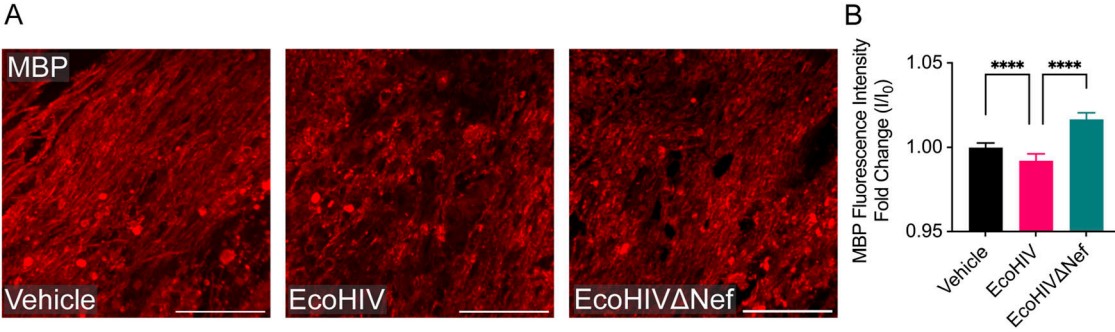

**Figure 5.   EcoHIV disrupts CNS myelin in the corpus callosum.**
Mice were intracranially injected as in Fig 1. At 5-wk post injection, brain tissue was processed for analysis by immunohistochemistry. **(A)** Representative fluorescence images of myelin basic protein (red) at the corpus callosum. **(B)** Quantification of MBP fluorescence intensity within the corpus callosum. n = 2–3 mice per group. Scale bar set to 50 μm. Results are presented as mean ± SEM and analyzed by Kruskal-Wallis rank sum test.

the disruption of transcriptional regulation of these lipid transport regulators.

Since Nef down-regulates ABCA1 function through a post-translational mechanism (38), we also examined ABCA1 protein abundance 5-wk post infection. A trend towards decreased ABCA1 protein abundance was observed in EcoHIV-infected mice relative to mock-infected and EcoHIVΔNef-infected mice (Fig 4C). Together, the data in Fig 4 suggest that transcriptional regulation of lipid transport regulators is disrupted in a viral mechanism independent of Nef, although post-transcriptional downmodulation of ABCA1 protein expression may be mediated by Nef in EcoHIV-infected mice, leading to persistently reduced abundance of ABCA1 in brain cells.

### Nef mediates disruption of myelin in EcoHIV-infected mice

Imaging studies have demonstrated corpus callosum thinning and loss of white matter integrity (39, 40), and postmortem transcriptome analyses have revealed alterations in myelin-related genes (19) in HIV-positive individuals on stable antiretroviral therapies. However, the role of Nef in this pathology remained unclear. Previous assessments of EcoHIV viral load reported a decline in HIV DNA after peaking at 10 d post infection and transient replication dropping drastically by 3 wk post infection; similarly,

induction of several inflammatory and immune activation genes at 15-d post infection was significantly reduced at 30-d post infection and typically comparable to saline-controls at 60-d post infection (31). Since we were interested in long-term effects of Nef on myelin, we examined myelin-binding protein (MBP) expression at 5-wk post-intracerebral infection via confocal microscopy. The myelin sheaths of EcoHIV-infected mice showed more visible disruption compared with vehicle- or EcoHIVΔNef-injected mice (Fig 5A). There was no significant difference in MBP fluorescence intensity between vehicle- and EcoHIVΔNef-injected mice; in contrast, mice injected with EcoHIV displayed a significant decrease in MBP fluorescence intensity (Fig 5B), suggesting Nef's involvement in persistent myelin perturbation in EcoHIV-infected mice.

### Nef contributes to hippocampal synaptodendritic loss in EcoHIV-infected mice

Previous studies have noted non-apoptotic hippocampal synaptodendritic injury in EcoHIV-infected mice, associated with neuronal dysfunction and neurocognitive impairment (28, 31). To investigate whether Nef contributes to this hippocampal neuronal injury, we examined MAP2 expression, a marker of neuronal axons and dendrites, at 5 wk post-intracerebral injection of EcoHIV or EcoHIVΔNef (Fig 6A). Consistent with earlier findings, EcoHIV-

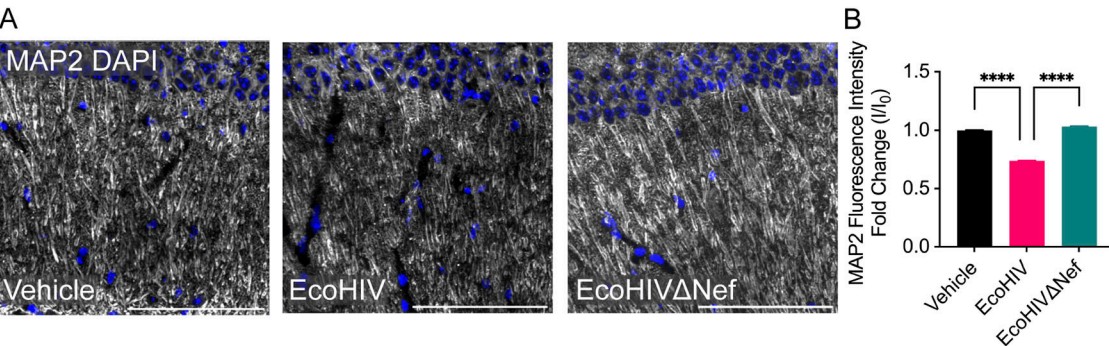

**Figure 6. Nef contributes to hippocampal neuronal injury in EcoHIV-infected mice.**
Mice were intracranially injected as in Fig 1. At 5-wk post injection, brain tissue was processed for analysis by immunohistochemistry. **(A)** Representative fluorescence images of axons and dendrites (MAP2, white) and nuclei (DAPI, blue). Images are from hippocampus at CA1. **(B)** Quantification of MAP2 fluorescence intensity within the CA1 hippocampal region. n = 3–5 mice per group. Scale bar is 100 $\mu m$. Results are presented as mean ± SEM and analyzed by Kruskal-Wallis rank sum test.

infected mice displayed decreased MAP2 fluorescence intensity in the CA1 region of the hippocampus (30% decrease relative to mock infection), which was not observed in mice infected with Eco-HIVΔNef (Fig 6B). No alterations in DAPI+ staining were registered among the groups, suggesting that the observed variances in MAP2 were not correlated with neuronal death. These results suggest that Nef may contribute to neuronal injury induced by EcoHIV in the hippocampus.

## Discussion

In this study, we explored the role of Nef in neuroinflammation and brain pathology triggered by intracranial injection of EcoHIV. EcoHIV DNA, both Nef-positive and ΔNef, persisted in brain tissue for up to 16 wk post infection, providing a model to investigate chronic Nef effects in the CNS. Our findings align with previous research on EcoHIV showing HIV DNA in the brains of peripherally infected mice up to 94 d after infection (29), and in the brains of intracranially injected mice up to 60 d post infection (31). Although we observed a slight decrease in HIV DNA in EcoHIVΔNef-infected mice compared with those infected with EcoHIV, the difference was not statistically significant. This suggests that while Nef may positively influence viral replication in EcoHIV-infected mice, its contribution to viral infectivity and spread appears less pronounced compared with simian immunodeficiency virus-infected primates or HIV-infected humans (41). Several factors, such as the method of virus inoculation and the relatively short duration of our experimental procedures relative to natural HIV infection, may contribute to the observed efficient infection with Eco-HIVΔNef. In addition, EcoHIV infection does not involve CD4, eliminating the boost to HIV spread via Nef-mediated CD4 downmodulation (42). Moreover, the effect of Nef on murine counterparts of human genes, such as SERINC 5 and MHC I, downmodulation of which contributes to Nef-mediated stimulation of HIV replication, has not been confirmed. These factors may limit the effectiveness of immune responses, which are pivotal in suppressing the replication of Nef-deficient HIV (23, 43). A notable advantage of our model system is the absence of the typical concern associated with studies involving Nef in HIV infection, namely, the suboptimal replication of Nef-deficient viruses.

Our study reveals that mice subjected to intracranial injection of EcoHIV exhibit an increase in the number of IBA1+ microglia/macrophages at the injection site, coupled with the up-regulation of the microglial/macrophage marker AIF-1 (IBA1). Notably, this activation was attenuated in mice infected with Nef-deficient virus. Furthermore, the expression of monocyte-derived inflammatory markers such as IL-1β, IL-6, CCL2 (MCP-1), and IFN-γ also showed an up-regulation in a Nef-dependent manner. AIF-1 (IBA1) serves as a marker for microglia/macrophage activation, delineating a proliferative and vigilant phenotype (44). Likewise, IL-1β is considered a key mediator of the inflammatory response (45); although it is almost undetectable in the healthy brain, it is rapidly induced in microglia minutes after acute experimental injuries (46, 47). IL-1β acts primarily on microglia, astrocytes, and endothelial cells (47). In response, cells stimulated by IL-1β produce a multitude array of signaling molecules including pro-inflammatory cytokines, chemotactic chemokines, reactive oxygen species, nitric oxide, glutamate, cell surface adhesion molecules, and prostaglandins (reviewed in reference 48). Biomarkers of monocyte activation and neuroinflammation IL-1β, IL-6, and CCL2 (MCP-1), frequently elevated in the brains and cerebrospinal fluid of people with PLWH on suppressive ART (49, 50, 51), were significantly elevated in EcoHIV-infected animals compared with those infected with EcoHIVΔNef. These results are in line with previous studies that reported that 2 wk after intracranial infection with EcoHIV, mouse brains displayed increased activation and infiltration of IBA1+ microglia as well as up-regulation of inflammation-related genes associated with HAND (31). Increased CCL2 production may attract local microglia and facilitate monocyte migration across the blood-brain barrier, thereby perpetuating a self-sustaining vicious cycle. We have previously reported that Nef EVs induce long-term hyperreactivity of myeloid cells, which may contribute to the persistent inflammation and associated pathologies observed in stably suppressed PLWH on ART (52).

The expression of TREM2 significantly increased in both EcoHIV- and EcoHIVΔNef-injected mice. TREM2 is known as a regulator of chronic inflammation, promoting phagocytosis, and microglial proliferation (53, 54). Elevated soluble TREM2 levels are associated with cognitive dysfunction and neuroinflammation in HIV+ brains, suggesting its potential as a HAND biomarker (37, 55). Conversely,

membrane-bound TREM2 is decreased in HAND patients (55). Although further studies on TREM2 distribution are needed for a comprehensive understanding, our findings suggest that EcoHIV infection impacts the TREM2 pathway in a Nef-independent manner.

Astrogliosis is a common finding in the HIV-positive brains and is associated with both astrocytic activation and apoptosis (56). Previous examination into the effects of HIV and Nef on GFAP expression has yielded conflicting results: some studies reported activation of astrocytes with increased GFAP expression (57), although other studies showed astrocyte loss and decreased GFAP expression after exposure to Nef in vitro (34), highlighting the heterogeneity and differential responses of astrocyte populations. In the context of EcoHIV infection, immunohistochemical analysis showed increased GFAP expression in the caudoputamen region in the vicinity of the injection at 2-wk post infection (33). We assessed the effect of EcoHIV on white matter astrocytes and observed increased GFAP expression in the corpus callosum at the site of injection; GFAP expression was further increased in mice infected with EcoHIVΔNef compared with both EcoHIV- and mock-infected mice. These results suggest that EcoHIV activates astrocytes independent of Nef, likely due to actions by other viral proteins such as Tat (58). Simultaneously, Nef may exert an inhibitory effect on GFAP expression in white matter. Mechanisms of this possible inhibitory effect remain to be investigated. Given that elevated expression of GFAP is a hallmark of reactive gliosis, the reduction in GFAP expression observed in our study may signify a specific influence of Nef on astrocytic function, potentially contributing to astrocyte dysfunction during astrogliosis.

We have previously shown that Nef potentiates inflammatory responses by post-transcriptionally downmodulating ABCA1 protein abundance, resulting in accumulation of intracellular cholesterol and increased abundance of the lipid rafts (5, 6). In this study, we observed only a trend towards decreased ABCA1 abundance in mice infected with Nef-positive EcoHIV. The small magnitude of the effect may be due to low level of infection in this model. Both EcoHIV- and EcoHIVΔNef-infected mice displayed down-regulation of ABCA1 RNA. This effect has not been reported in previous studies, and likely potentiates the post-translation effect of Nef (38), contributing to ABCA1 downmodulation.

We present the first examination of myelin impairment in EcoHIV-infected mice. Myelin sheaths were notably disrupted within the corpus callosum at the injection site compared with mock- and EcoHIVΔNef-injected animals. EcoHIV-infected mice exhibited significantly decreased MBP fluorescence intensity compared with both mock- and EcoHIVΔNef-infected mice, suggesting a role for Nef in EcoHIV-mediated myelin disruption. Moreover, there were no differences in MBP fluorescence intensity between mock- and EcoHIVΔNef-injected animals. The loss of myelin may reflect a cytotoxic effect of Nef on oligodendrocytes (35). Unfortunately, our attempts to quantify oligodendrocyte numbers did not provide a definitive answer. Nevertheless, the effect of Nef on myelin correlated with an increase in IBA1+ microglia/macrophages at the injection site, indicating that chronic inflammation may contribute to the observed myelin impairment in EcoHIV-infected mice. Neuroinflammation can either promote or reflect active demyelination, and the established link between

oligodendrocyte death and demyelinating diseases supports this view (reviewed in reference 20). Oligodendrocyte stress closely follows microglial activation (59), which release cytokines and other cytotoxic factors triggering oligodendrocyte apoptosis (60, 61). Future studies will use advanced technology, including high-resolution microscopy, to analyze the effects of Nef on myelin ultrastructure in EcoHIV-infected animals.

Previous studies have indicated that EcoHIV-infected mice exhibit non-apoptotic hippocampal synaptodendritic injury, linked with neuronal dysfunction and neurocognitive impairment (28, 31). In addition, evidence from other studies suggests the toxic effects of Nef and Nef EVs on neurons (16, 17, 62). Our findings demonstrate that the effects on hippocampal MAP2 immunostaining observed in EcoHIV-infected mice are significantly reduced in mice infected with EcoHIVΔNef, implicating Nef in hippocampal synaptodendritic injury. Future studies will examine Nef's contribution to hippocampal learning and memory deficits observed in EcoHIV-infected mice.

Several limitations of this study should be acknowledged. Foremost is the small number of animals analyzed, stemming from challenges in establishing robust brain infection with EcoHIV. Consequently, this report provides only a preliminary assessment of the effects of Nef in the brain. Secondly, there are inherent limitations of the EcoHIV model itself. The excision of HIV gp120 and its replacement with murine leukemia virus gp80 precludes the analysis of gp120's reported neurotoxic effects (18). Lastly, the functional activity of the HIV protein Tat, implicated in HIV-associated neurotoxicity (63), is inherently impaired in murine cells (64, 65).

In conclusion, our study offers the initial characterization of Nef's impact on HIV-associated neuropathology in EcoHIV-infected mice. Nef emerges as a significant contributor to heightened neuroinflammation, leading to disruptions in white matter astrocytes, myelin impairment, and synaptodendritic injury within the hippocampus. The Nef-mediated down-regulation of ABCA1, along with subsequent disturbances in cholesterol homeostasis, likely contributes to the observed neuropathologies in EcoHIV-infected mice. Furthermore, the exploration of Nef's effects within the EcoHIV model of HIV-associated neuropathology holds promise for uncovering novel pathogenic mechanisms and identifying potential therapeutic targets crucial for managing this serious comorbidity. These results build upon earlier investigations demonstrating the elevation of several pro-inflammatory factors within the brains of EcoHIV-infected mice (31), and extend to microglial cells our previous observations in MDM about the pro-inflammatory activity of Nef (5, 6).

# Materials and Methods

### Mice

C57BL/6J mice (Jackson Laboratory) were maintained under standard mouse husbandry conditions including a 12:12 light:dark cycle and ad libitum access to food and water. All experimental procedures were approved by the George Washington University

Institutional Animal Care and Use Committee (protocol A2020-40, approved on 15 December 2020).

### Infectious virus stocks

The viruses EcoHIV/NDK and EcoHIV/NDK-ΔNef, referred to as EcoHIV and EcoHIVΔNef, are shown in Fig S1A. Virus stocks were produced as described previously (66). Briefly, HEK293T cells were cultured in T75 cc flasks at 70% confluency and transfected with 10 μg of plasmid DNA encoding viruses EcoHIV or EcoHIVΔNef using Metafectene (#T020-5.0; Biontex) per the manufacturer's instructions. Cells and culture supernatants were collected 72 h after transfection. Cells were lysed and analyzed by Western blot (see below) using the mixture of 3D12 and JR6 antibodies from Abcam and HRP-conjugated secondary anti-mouse antibody (Bio-Techne) (Fig S1B). Supernatants were centrifuged at 500$g$ for 10 min to remove cells, then at 1,500$g$ for 30 min to remove cell debris. Supernatants were collected and concentrated by ultracentrifugation at 72,173$g$ in the Optima XPN -100 Ultracentrifuge using a TI-100 rotor. Viral pellets were washed and resuspended in saline for injection and p24 concentration was measured using the Alliance HIV-1 Elisa Kit (Perkin-Elmer).

### Infection of mice

Mice were infected as previously described (31). Briefly, 20–24-wk-old male and female mice were infected by stereotaxic injection into the caudoputamen region of the right hemisphere (anterior-posterior 0 mm, medial-lateral 2 mm, dorsal-ventral 3 mm). Accuracy of coordinates was confirmed with injection of Evans blue dye prior to conducting experiments. After cleaning each mouse's head with betadine and making a small incision with a surgical scalpel, a burr hole was drilled into the skull at 2 mm lateral to bregma (Fig 1A). A Hamilton syringe needle was lowered to a depth of 3 mm into the drilled hole then retracted slightly. 10 μl of sterile saline (mock infection) or virus containing 2 × 10$^6$ pg of p24 of EcoHIV or EcoHIVΔNef was injected at a rate of 0.5 μl/min. At 1-, 2-, or 5-wk post injection, brains were collected and prepared for analysis of viral burden, cellular gene expression, and microscopy. Areas of interest included the ipsilateral (injected) caudoputamen region, corpus callosum, and hippocampus and contralateral corresponding areas as controls. In separate studies, 10 μl of sterile saline (vehicle) or virus containing 1 × 10$^6$ pg of p24 of EcoHIV or EcoHIVΔNef was injected at a rate of 0.5 μl/min, and brain tissue was collected at 5-wk post injection for analysis of ABCA1 protein expression or at 16-wk post injection for analysis of viral burden in the ipsilateral (injected) and contralateral hemispheres.

### Measurement of viral burden and cellular gene expression

Brain tissue was analyzed for viral burden and cellular gene expression as described previously (31). Briefly, mice were transcardially perfused with chilled 1xPBS, and brain tissue was collected. For brain tissue collected at 16-wk post injection, 3 mm coronal sections containing the injection site were obtained using a mouse brain matrix, and sections were cut midsagitally, placed in RNA*later* Stabilization Solution (AM7020; Invitrogen), and stored at

4°C. For brain tissue collected at 1- or 2-wk post injection, sections were flash frozen and stored at −80°C.

Frozen sections (~30 mg each) were grounded into powder using pellet pestles in 1.5 ml tubes (#12-141-368; Thermo Fisher Scientific). Genomic DNA was isolated from samples using the QIAamp Fast DNA Tissue Kit (#51404; QIAGEN), and total RNA was isolated using the RNeasy Lipid Tissue Mini Kit (#74804; QIAGEN). EcoHIV *gag* transcript was detected by real-time quantitative PCR (qPCR) using the iQ SYBR Green Supermix (#1708880; Bio-Rad) using custom primers for EcoHIV *gag* (forward primer: 5′-TGGGACCACAGGCTA-CACTAGA-3′ and reverse primer: 5′-CAGCCAAAACTCTTGCTTTATGG-3′, Integrated DNA Technologies). EcoHIV *gag* DNA copies were quantified against a standard curve using graded numbers of EcoHIV plasmid. DNA copy number was calculated using the conversion factor specific for C57BL/6J mice of 6.25 pg DNA per diploid cell (21) and normalized to 1 million cells.

For qRT-PCR of RNA transcript expression, 0.5 μg of RNA was reverse transcribed into cDNA using the iScriptTM cDNA synthesis kit (#1708891; Bio-Rad) under the following conditions: 25°C for 5 min, 46°C for 20 min, and 95°C for 1 min. Real-time PCR assay was performed using the iQ SYBR Green Supermix and gene-specific primers (Table S1). The PCR reactions were performed on a Bio-Rad CFX96 Touch Real-Time PCR Detection System. Data were analyzed in the Bio-Rad CFX Manager Software 3.0 according to the manufacturer's instructions. Samples for qRT-PCR were run in duplicate under the following conditions: 95°C for 3 min, 40 cycles of 95°C for 15 s and 60°C for 1 min, and 65 to 95°C in increments of 0.5°C at 5 s/step. Relative gene expression was normalized to the housekeeping gene $β$-actin and quantified by the comparative quantification cycle (ΔΔCq) calculation method (Horizon Discovery tech note). Briefly, given the Cq values for the reference housekeeping (REF) and target (TARG) genes, ΔΔCq was calculated as follows (Table S2): (i) Data normalization to REF gene, (ii) exponential transformation, (iii) averaging the replicates and calculation of SD for each treatment condition, and (iv) normalization to treatment control.

### Western blot analysis of cells and tissue

Automated capillary western immunoblots were performed using the ProteinSimple Jess Simple Western system (Bio-Techne) according to the manufacturer's instructions as described previously (67). Brain samples or transfected HEK293T cells were homogenized in NET buffer (NP-40 lysis buffer: 150 mM NaCl, 1% NP-40, 50 mM Tris–HCl at pH 8.0) containing protease and phosphatase inhibitors (#78420; Thermo Fisher Scientific). Then cell lysates were centrifuged at 700$g$ for 5 min. Four microliters of cleared cell lysate (1 μg/μl protein) were mixed with 1 μl of 5x master mix containing fluorescent molecular weight markers, 200 mM dithiothreitol, KPL blocking buffer (#5920-0004; SeraCare Life Sciences), total protein quantification reagent, mouse monoclonal anti-Nef antibodies (mixture of 3D12 diluted 1:40 and JR6 diluted 1:80, #ab42355 and #ab42358, respectively; Abcam) and HRP-conjugated secondary anti-mouse (#042-205; Bio-Techne) (for Nef detection), and chemiluminescent substrate were loaded on the instrument. For ABCA1 analysis, rabbit polyclonal anti-ABCA1 (1:40; #ab63918; Abcam) and secondary HRP-

conjugated anti-rabbit antibody (#042-206; Bio-Techne) replaced the anti-Nef antibody. Lysates were incubated with 5x master mix at 37°C for 15 min.

After loading, the separation, electrophoresis, total protein quantification, and immunodetection steps were conducted in the fully automated Jess system as described (35). Digital image of chemiluminescence of the capillary was captured with Compass Simple Western software (version 5.1.0, Protein Simple) that automatically calculated heights (chemiluminescence intensity), area, and signal/noise ratio. Results were visualized as electropherograms representing the peak of chemiluminescence intensity and as lane view from the signal of chemiluminescence detected in the capillary.

### Immunohistochemistry analysis of brain tissue

At 5-wk post injection, mice were transcardially perfused with 4% paraformaldehyde (#15710; EMS). Brain tissue was collected and post-fixed overnight at 4°C, cryopreserved in 30% sucrose, embedded in Tissue-Tek O.C.T. Compound (Sakura), and frozen at –80°C. 10 $\mu m$ coronal sections were obtained using a Leica CM1950 cryostat microtome and sections were dried overnight at room temperature before storage at -20°C. To process tissue for immunohistochemistry as described previously (68), frozen sections were dried at room temperature for 1 h, then rehydrated in 1xPBS for 5 min. All samples were then incubated for 1 h at room temperature in 10% NGS, 0.3% Triton X-100 (Sigma-Aldrich) in 1xPBS, followed by primary antibodies overnight at 4°C and secondary antibodies for 1 h at room temperature. All antibodies and dilutions are listed in Table S3. Sections were stained with DAPI (1:3,000; #46190; Thermo Fisher Scientific) and mounted in ProLong Gold Antifade Mountant (#P36934; Invitrogen).

### Microscopy of brain samples

Confocal microscopy was performed at the George Washington Nanofabrication and Imaging Center. Confocal 3D-images of GFAP, MBP, and MAP2 immunostaining were captured using the Zeiss Cell Observer Z1 spinning disk confocal microscope (Carl Zeiss, Inc.) equipped with ASIMS-2000 (Applied Scientific Instrumentation) scanning stage with z-galvo motor, and Yokogawa CSU-X1 spinning disk. Zen Blue software (Carl Zeiss, Inc.) was used with 10x and 40x objectives to acquire 3D-tile images. Exposure settings were maintained between all treatment conditions. Confocal images were analyzed with Imaris AI Microscopy Image Analysis Software version 10.0 (Oxford Instruments). For GFAP and MBP analysis, images were acquired of the corpus callosum at the injection site and stitched as previously described (69). New surfaces per channel were created for fluorescence intensity quantification to segment and quantify the labeling (Imaris 9.5). We performed background subtraction to define the background at each voxel, and then a baseline subtraction of this variable was performed using Gaussian filtering. Thresholds were selected by visual inspections although filtering (number of voxels) was applied to create each surface model. Surface editing was performed to delete the remaining background labeling, and then a mask channel was created to extract the intensity information for their corresponding Alexa Fluor channel. In each new mask channel, the statistic values per surface (intensity mean) and the intensity histograms were extracted automatically. 400 × 500 $\mu m$ ROIs were analyzed for GFAP fluorescence intensity. 300 × 300 $\mu m$ ROIs were

analyzed for MBP fluorescence intensity using a seed point diameter of 0.5 $\mu m$. For analysis of MAP2, images were acquired in the CA1 hippocampal region and stitched. For each tile scan, three 100 × 100 $\mu m$ ROIs were analyzed for MAP2 fluorescence intensity using a seed point diameter of 0.5 $\mu m$. Fluorescence images of IBA1 immunostaining were acquired using a Leica DM 5500 upright microscope with a Hamamatsu ORCA-R2 camera. 20x images were acquired of the basal ganglia at the injection site and analyzed in ImageJ (NIH) for staining area and cell counts.

### Statistical analysis

Statistical analyses were performed using GraphPad Prism 10 Software. A Shapiro-Wilkinson normality test was conducted, followed by a one-way analysis of variance (ANOVA) with Tukey's post hoc test or Kruskal-Wallis with Dunn's post hoc test where appropriate. Results are presented as mean ± SEM or mean ± SD. $P$-value representations are indicated (*$P < 0.05$; **$P < 0.01$; ***$P < 0.001$, ****$P < 0.0001$).

# Data Availability

All data generated or analyzed during this study are included in this published article and its supplementary information files.

# Supplementary Information

# Acknowledgements

This study was supported by AHA grant 20PRE35080036 (JK Schenck) and NIH grants to MI Bukrinsky R01 NS124477 and P30 AI117970. We thank Dr. Volsky for a kind gift of EcoHIV and EcoHIVΔNef viruses. We also thank the GWU Imaging and Nanotechnology Center for guiding the confocal imaging analysis.

### Author Contributions

JK Schenck: investigation, methodology, and writing—original draft.
C Clarkson-Paredes: data curation, investigation, visualization, and methodology.
T Pushkarsky: investigation.
Y Wang: investigation.
RH Miller: conceptualization, supervision, methodology, and writing—review and editing.
MI Bukrinsky: conceptualization, data curation, supervision, funding acquisition, methodology, project administration, and writing—review and editing.

### Conflict of Interest Statement

The authors declare that they have no conflict of interest.

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
