## [Reviewer comments · Life Science Alliance]

Life Science Alliance

Nef Mediates Neuroimmune Response, Myelin Impairment, and Neuronal Injury in EcoHIV-Infected Mice

Jessica Schenck, Cheryl Clarkson-Paredes, Tatiana Pushkarsky, Yongsen Wang, Robert Miller, and Michael Bukrinsky
DOI: <https://doi.org/10.26508/lsa.202402879>

Corresponding author(s): Michael Bukrinsky, George Washington University

Review Timeline:

Submission Date:	2024-06-07
Editorial Decision:	2024-08-12
Revision Received:	2024-10-31
Editorial Decision:	2024-11-01
Revision Received:	2024-11-02
Accepted:	2024-11-04

Transaction Report:

August 12, 2024

Re: Life Science Alliance manuscript #LSA-2024-02879-T

Prof. Michael Bukrinsky
GWU SMHS
Microbiology, Immunology and Tropical Medicine
2300 Eye Street NW
Ross Hall 624
Washington, District of Columbia 20037

Dear Dr. Bukrinsky,

Thank you for submitting your manuscript entitled "Nef mediates a neuroimmune response, myelin impairment, and hippocampal neuronal injury in EcoHIV-infected mice." to Life Science Alliance. The manuscript was assessed by expert reviewers, whose comments are appended to this letter. We invite you to submit a revised manuscript addressing the Reviewer comments.

Thank you for this interesting contribution to Life Science Alliance. We are looking forward to receiving your revised manuscript.

Sincerely,

B. MANUSCRIPT ORGANIZATION AND FORMATTING:

Reviewer #1 (Comments to the Authors (Required)):

Authors used a murine model infected with Nef-expressing (EcoHIV) and Nef-deficient (EcoHIV Δ Nef) murine HIV. They showed increased neuroinflammation, reduced myelin and neuronal integrity in EcoHIV-infected compared to EcoHIV Δ Nef-infected brains. Both viruses induced astrogliosis in infected brains, with stronger activation of GFAP in mice infected with the Nef-deficient virus. Authors conclude that Nef contributes to neuroinflammation primarily by targeting microglial cells, while other factors may regulate astrogliosis. These findings indicate that Nef may be a significant contributor to neuronal injury in EcoHIV-infected mice. Study contributes to our understanding of Nef toxicity in the CNS and effects on neuroinflammation, myelin injury and neuronal damage. A number of issues exist in this study (some acknowledged by investigators).

1. Neuroinflammation (genes of pro-inflammatory factors) and viral load were investigated at 2 weeks, while immunohistochemistry was done at 5 weeks post infection. There is no explanation why studies that should be performed at the same time point were done 3 weeks apart.
2. Myelin studies using MPB should be complemented by immunostains for oligodendrocyte marker.
3. Increased GFAP reaction in EcoHIV Δ Nef group should be better explained.
4. Authors used term 'infiltration' for Iba1 staining. What evidence they have that part of Iba1+ cells came from periphery?
5. MAP2 staining should be complemented by NeuN immunostain.

Reviewer #2 (Comments to the Authors (Required)):

Manuscript# LSA-2024-02879-T :

The article 'Nef mediates a neuroimmune response, myelin impairment, and hippocampal neuronal injury in EcoHIV-infected mice' by Schenck J.K. et al. demonstrated Nef protein roles in neuronal injury using the EcoHIV system. The research strategy was sound. However, some evidence does not support the hypothesis, and some data need to be improved.

Major points

Line 121-125: Please include the qPCR probe information.

Fig.3D: The vehicle fluorescence photo needs to be replaced with a better-quality photo. It does not show the difference clearly.

Fig.5: Both (A) and (B) data are not convincing. In (B), the fold change of the image is shown in (A). The present fold change number looks like vehicle 1, EcoHIV 0.98, and EcoHIV Δ Nef 1.02 or 1.03. Is this small difference significantly different?

Fig.6A: For the vehicle photo, please present the photo with a similar number of DAPI. The current photo is shifted compared to EcoHIV and EcoHIV Δ Nef and does not show the DAPI enough. It looks like the same area, but the photo for the vehicle seems shifted. Also, to support hippocampal neuronal injury, it will be better to show how cerebral neurons look.

Supplemental Fig.S1 B: This Western blot membrane has been cut into strips, which was unnecessary. This experiment used only one antibody (ARP-3689) to confirm nef expression, which means there is no reason to cut the membrane into strips. I suggest showing a whole membrane as one piece to make the data more convincing. To improve Western blot data, also suggest adding housekeeping protein data.

Minor points

Line 87: 10 mg of plasmid DNA for T75 with HEK293T cells sounds like a very high concentration. This could be a typo so please check again.

Line 148: 1% NP-40 has been duplicated in the buffer composition.

Line 209: The symbols for p values 0.001 and 0.0001 are the same. Please differentiate from each other.

Reviewer #3 (Comments to the Authors (Required)):

The manuscript presents an preliminary report wherein Nef was tested for its role in neuroinflammation including astrogliosis, myeloid cell migration, and myelin impairment in an EcoHIV model of human HIV.

The results generally support the conclusions. addressing a the following considerations will provide support or context.

Nef expression in situ:

Did you look for Nef protein or mRNA to verify that it was expressed? Nef antibodies are notoriously challenging - some have found they work better with IHC than IF. If protein data is not available, at least Nef RT-PCR data would be helpful to support statements attributing an effect of Nef.

Was Nef antibody from the NIH AIDS reagent program? which antibody and what dilution? This information is missing from S3.

The results show myeloid cell infiltration and upregulation of CCL2 (fig 3 and fig 2D) with similar pattern between EcoHIV and deltaNef. Some discussion is warranted on the link, in if still possible (samples available), examination of blood brain barrier disruption in the vehicle, EcoHIV and deltaNef would suggest a mechanism/source of the increased IBA1 signal.

Please include the stereotaxic coordinates and provide information on how the accuracy of the infusion site was confirmed? Also address which cells were infected.

these modifacaitons are considered minor but would support the findings and better point to the role of Nef as well as strengthen the discussion.

Referee cross comments - none.

Reviewer 1

- 1. Neuroinflammation (genes of pro-inflammatory factors) and viral load were investigated at 2 weeks, while immunohistochemistry was done at 5 weeks post infection. There is no explanation why studies that should be performed at the same time point were done 3 weeks apart.**

Response: We apologize that we did not clearly explain the logic behind the selected time points. The rationale for the different post injection intervals is that the 2 approaches were designed to address somewhat different questions. Previous assessments of EcoHIV viral load reported a mild to moderate decline in HIV DNA after peaking at 10 days post infection and transient replication dropping drastically by 3 weeks post-infection; similarly, induction of several inflammatory and immune activation genes at 15-days post-infection was significantly reduced at 30-days post-infection and typically comparable to saline-controls at 60-days post infection (PMID: 31266862). We were interested in investigating whether Nef contributes to the induction of pro-inflammatory genes as assayed at 2 weeks post injection and whether this has an effect on persistent glial dysfunction and myelin impairment as assayed at 5 weeks post injection. These results may help explain why neurocognitive impairments persist in HIV positive individuals with undetectable viral loads. This is now explained in the revised text (line 322).

- 2. Myelin studies using MPB should be complemented by immunostains for oligodendrocyte marker.**

Response: The goal of these studies was to determine whether there was a loss of myelin, and this is best demonstrated through the use of antibodies to MBP, the major myelin protein. We agree that the loss of myelin may reflect changes in oligodendrocytes or simply loss/disruption of myelin from existing cells. Attempts to quantify oligodendrocyte numbers did not provide a definitive answer and such studies will require the development of an in vitro assay that is beyond the scope of the current study. We have added a comment to this effect in the discussion on line 425.

- 3. Increased GFAP reaction in EcoHIV Δ Nef group should be better explained.**

Response: Observed inhibitory effect of Nef on GFAP expression remains to be investigated. We have added discussion of this issue on line 402.

- 4. Authors used term 'infiltration' for Iba1 staining. What evidence they have that part of Iba1+ cells came from periphery?**

Response: We agree with the reviewer and apologize for not being clear in our descriptions. Based on Iba1 staining we cannot distinguish between infiltrated and resident Iba1+ cells and have changed the text to remove the term "infiltration" replacing it with "increase" (lines 266, 427).

- 5. MAP2 staining should be complemented by NeuN immunostain.**

Response: We believe the reviewer is asking whether there was a significant loss of neuronal cells in the hippocampus that accompanied EcoHIV infection. Consistent with previous studies, while we observed a reduction in MAP2 expression, we did not see any noticeable change in total cell numbers in the CA1 region suggesting a response in dendritic retraction/injury rather than neuronal loss. This is now discussed on line 339.

Reviewer 2

1. Line 121-125: Please include the qPCR probe information.

Response: Our experiments utilize Bio-Rad's iQ SYBR® Green Supermix, a type of intercalating dye that does not require a probe design. All primers used for PCR analysis are listed in Supplemental Table S1.

2. Fig. 3D: The vehicle fluorescence photo needs to be replaced with a better-quality photo. It does not show the difference clearly.

Response: We have updated Fig. 3D to include a higher magnification inset of GFAP staining within the corpus callosum.

3. Fig. 5: Both (A) and (B) data are not convincing. In (B), the fold change of the image is shown in (A). The presented fold change number looks like vehicle 1, EcoHIV 0.98, and EcoHIVΔNef 1.02 or 1.03. Is this small difference significantly different?

Response: For MBP analysis, we segmented voxels using a seed point diameter of 0.5 μm (line 197). This generated a large number of data points per sample, allowing us to detect small but significant differences between groups. We have adjusted the graph axis to better show these differences.

4. Fig. 6A: For the vehicle photo, please present the photo with a similar number of DAPI. The current photo is shifted compared to EcoHIV and EcoHIVΔNef and does not show the DAPI enough. It looks like the same area, but the photo for the vehicle seems shifted. Also, to support hippocampal neuronal injury, it will be better to show how cerebral neurons look.

Response: We have replaced the figure to address the issue raised by the Reviewer. Unfortunately, we do not have samples left to examine cerebral neurons.

5. Supplemental Fig. S1B: This Western blot membrane has been cut into strips, which was unnecessary. This experiment used only one antibody (ARP-3689) to confirm nef expression, which means there is no reason to cut the membrane into strips. I suggest showing a whole membrane as one piece to make the data more convincing. To improve Western blot data, also suggest adding housekeeping protein data.

Response: We have reanalyzed these samples using a mixture of two anti-Nef antibodies. We used the automated capillary Western blot instrument from BioTechne for this analysis. This

instrument performs analysis of the protein of interest (Nef in our case) and total protein (as a loading control) in the same capillary. Both are now presented. Results from analysis on this instrument are produced as individual lanes representing each capillary.

- 6. Line 87: 10 mg of plasmid DNA for T75 with HEK293T cells sounds like a very high concentration. This could be a typo so please check again.**

Response: We apologize for the typo, it should be 10 µg. We corrected this typo.

- 7. Line 148: 1% NP-40 has been duplicated in the buffer composition.**

Response: We have corrected this typo.

- 8. Line 209: The symbols for p values 0.001 and 0.0001 are the same. Please differentiate from each other.**

Response: We have corrected this typo.

Reviewer 3

- 1. Nef expression in situ: Did you look for Nef protein or mRNA to verify that it was expressed? Nef antibodies are notoriously challenging - some have found they work better with IHC than IF. If protein data is not available, at least Nef RT-PCR data would be helpful to support statements attributing an effect of Nef.**

Response: Unfortunately, samples are no longer available for protein or RNA analysis. However, initial experiments clearly demonstrated viral RNA expression (Fig. 1D). Therefore, given the presence of the Nef gene in the wild-type virus and its absence in the Δ Nef virus (Supplemental Fig. S1), it is reasonable to assume that Nef was expressed in the brains of infected mice.

- 2. Was Nef antibody from the NIH AIDS reagent program? which antibody and what dilution? This information is missing from S3.**

Response: Information about the anti-Nef antibodies has been added on lines 90 and 152, as well as in the legend to Supplemental Fig. S1.

- 3. The results show myeloid cell infiltration and upregulation of CCL2 (fig 3 and fig 2D) with similar pattern between EcoHIV and deltaNef. Some discussion is warranted on the link, in if still possible (samples available), examination of blood brain barrier disruption in the vehicle, EcoHIV and deltaNef would suggest a mechanism/source of the increased IBA1 signal.**

Response: Regrettably, the samples are no longer available for further examination of the blood-brain barrier (BBB). However, we have included a discussion regarding the potential relationship between increased levels of CCL2 and the observed elevation of IBA1 on line 378.

4. Please include the stereotaxic coordinates and provide information on how the accuracy of the infusion site was confirmed? Also address which cells were infected.

***Response:** Stereotaxic coordinates and details regarding accuracy of injection were added (line 97). We have not verified the type of infected cells, but previous reports have indicated that EcoHIV is localized to IBA1+ microglia/macrophages in the brain in vivo (PMID: 29879225) and in mixed brain cultures in vitro (PMID: 38793575). This is now mentioned on line 216.*

November 1, 2024

RE: Life Science Alliance Manuscript #LSA-2024-02879-TR

Prof. Michael Bukrinsky
George Washington University
Microbiology, Immunology and Tropical Medicine
2300 Eye Street NW
Ross Hall 624
Washington, District of Columbia 20037

Dear Dr. Bukrinsky,

Thank you for submitting your revised manuscript entitled "Nef Mediates Neuroimmune Response, Myelin Impairment, and Neuronal Injury in EcoHIV-Infected Mice". We would be happy to publish your paper in Life Science Alliance pending final revisions necessary to meet our formatting guidelines.

- please be sure that the authorship listing and order is correct
- please remove a legend from Figure S1 and add it to the main manuscript text after the legends for the main figures
- please remove figures from the main manuscript text and leave them uploaded separately
- please consult our manuscript preparation guidelines <https://www.life-science-alliance.org/manuscript-prep> and make sure your manuscript sections are in the correct order
- please add your main and supplementary figure legends to the main manuscript text after the References section
- please incorporate any points from the Conclusion section into the Discussion; we only allow a Discussion section
- please use the [10 author names et al.] format in your references (i.e., limit the author names to the first 10)

A. FINAL FILES:

B. MANUSCRIPT ORGANIZATION AND FORMATTING:

Sincerely,

November 4, 2024

RE: Life Science Alliance Manuscript #LSA-2024-02879-TRR

Prof. Michael Bukrinsky
George Washington University
Microbiology, Immunology and Tropical Medicine
2300 Eye Street NW
Ross Hall 624
Washington, District of Columbia 20037

Dear Dr. Bukrinsky,

Thank you for submitting your Research Article entitled "Nef Mediates Neuroimmune Response, Myelin Impairment, and Neuronal Injury in EcoHIV-Infected Mice". It is a pleasure to let you know that your manuscript is now accepted for publication in Life Science Alliance. Congratulations on this interesting work.

DISTRIBUTION OF MATERIALS:

Again, congratulations on a very nice paper. I hope you found the review process to be constructive and are pleased with how the manuscript was handled editorially. We look forward to future exciting submissions from your lab.

Sincerely,
